# DocTrack: A Visually-Rich Document Dataset Really Aligned with Human Eye Movement for Machine Reading

**Hao Wang[1], Qingxuan Wang[1], Yue Li[1], Changqing Wang[1], Chenhui Chu[2]\* and Rui Wang[3]**

[1]School of Computer Engineering and Science, Shanghai University, China
[2]Graduate School of Informatics, Kyoto University, Japan
[3]Department of Computer Science and Engineering, Shanghai Jiao Tong University, China
{wang-hao, wqx, ly_ces, wcqing}@shu.edu.cn
chu@i.kyoto-u.ac.jp, wangrui12@sjtu.edu.cn

## Abstract

The use of visually-rich documents (VRDs) in various fields has created a demand for Document AI models that can read and comprehend documents like humans, which requires the overcoming of technical, linguistic, and cognitive barriers. Unfortunately, the lack of appropriate datasets has significantly hindered advancements in the field. To address this issue, we introduce DOCTRACK, a VRD dataset really aligned with human eye-movement information using eye-tracking technology. This dataset can be used to investigate the challenges mentioned above. Additionally, we explore the impact of human reading order on document understanding tasks and examine what would happen if a machine reads in the same order as a human. Our results suggest that although Document AI models have made significant progress, they still have a long way to go before they can read VRDs as accurately, continuously, and flexibly as humans do. These findings have potential implications for future research and development of Document AI models. The data is available at https://github.com/hint-lab/doctrack.

## 1 Introduction

With the continuous development of information technology, our access to information is becoming increasingly diverse. Among the various formats, the proportion of visual information in daily documents such as tables, graphs, diagrams, etc., is on the rise (Ceci et al., 2007; Jaume et al., 2019). Therefore, effectively utilizing such visual information has become a hot research topic in the NLP research community and introduces a new challenge of understanding Visually-Rich Documents (VRDs) (Liu et al., 2019; Yu et al., 2021), which are documents that contain substantial visual components.

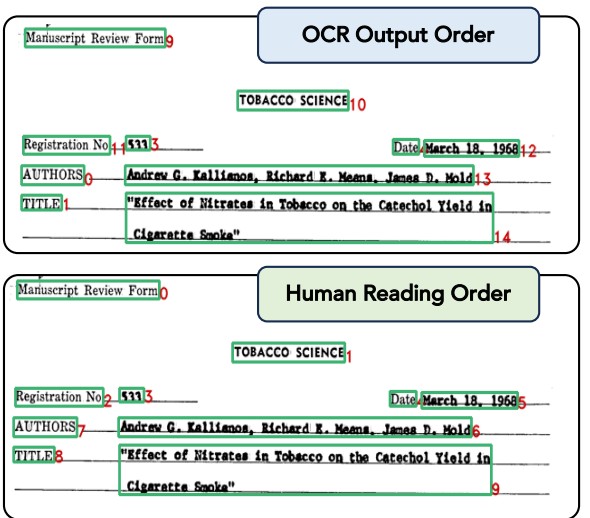

Figure 1: Case comparison of the default serialized input order output by the OCR engine with the real human reading order. It clearly reveals the significant differences in processing behavior between machines and humans. The red numbers indicate the ordinal numbers in the reading sequence. For best visibility, it is recommended to zoom in on the results.

Identifying and understanding VRDs is a time-consuming and laborious task due to their diversity and complexity (Wang et al., 2021a). Textual information alone is insufficient for extracting key information from diverse document types, necessitating a multimodal approach that considers the consistency and correlation of multiple modalities, including text, visual, and layout, through joint modeling. Examples of modern document AI models for VRD understanding include the LayoutLM series models (Xu et al., 2020, 2021; Huang et al., 2022) and StructText series models (Li et al., 2021; Yu et al., 2023a).

While these models can obtain fine-grained multimodal document representations and achieve promising results in downstream VRD (VRD) understanding tasks, they lack the ability to generate a serialized input order from a given document

---
\*Corresponding author

that fits into the Transformer architecture. As a result, they typically utilize simple rules, such as left-to-right or top-to-bottom, or directly use the input order generated by the OCR tool in the previous step (Lee et al., 2021) to serialize inputs. However, these input orders are quite different from the reading order that humans are used to (See Figure 1 for an example). The different input orders can significantly affect the performance of document AI models on the downstream document understanding tasks, which is often overlooked.

To address this issue, Wang et al. (2021b) construct the ReadingBank dataset containing the local priority order using the order of the XML source code of Word documents. However, it remains questionable whether this reading order is consistent with the actual human reading order and whether it is actually beneficial for machine comprehension tasks. Therefore, state-of-the-art Document AI models lack a deep understanding of spatial relationships and document structure, resulting in limited performance when dealing with VRDs, e.g., forms and infographics.

To this end, we propose DOCTRACK, a benchmark dataset containing various types of real-world documents aligned with human eye-movement information. Specifically, we propose a *preordering* pipeline to integrate human reading order within modern document AI models. The integration process occurs during inputting document contents after OCR parsing but before feeding them into the downstream task. We also explore different approaches to generate human-like reading orders, including default OCR tools, *Z*-pattern, simple rules, and AI models that utilize multimodal information. Our objective is to evaluate the impact of different reading orders on downstream document comprehension tasks and identify the most effective reading order for existing multimodal document AI models. This provides insight into the similarities and differences between human and machine reading patterns.

In summary, this paper makes three contributions:

1. We construct a benchmark dataset, namely DOCTRACK, aligned with human eye movement information. To our knowledge, DOCTRACK is the first human-annotated benchmark dataset for the purpose of research on VRD reading order generation.

2. We investigate different human-like reading order generation methods, which refer to techniques for generating machine reading orders that mimic human reading patterns in VRDs. We explore various techniques for generating these reading orders and propose a practical preordering pipeline that leverages these generated reading orders to improve document understanding tasks.

3. We conduct both intrinsic and extrinsic evaluations to analyze the performance of the human-like reading order generation model and measure its impact on downstream tasks. The observations suggest that human reading order may not be suitable for reading VRDs.

## 2 Human Reading Order

The human reading order plays a significant role in the comprehension of VRDs. Human reading order refers to the direction and sequential order in which people scan documents as they read the texts. Generally, people read left-to-right or top-to-bottom to comprehend the text and obtain information. This order is also related to the way the text is written in most languages and scripts, including English and Modern Chinese. However, other reading order patterns exist. For example, traditional Chinese and Japanese are usually written vertically from right to left, while Arabic is written horizontally from right to left (Rayner, 1998). As a result, people naturally scan their eyes in this order while reading (Henderson and Ferreira, 1993).

Although the reading order of text generally follows a relatively fixed pattern, VRDs usually present information in a mixture of modalities (e.g., text, images, graphics, etc.) with a complex two-dimensional layout and semantic structure. Readers typically choose the appropriate reading order by considering the spatial structure of text content, image information, and the typographic position of the document in combination. This natural temporal processing can help readers better understand the content and intent of the document and establish connections between different modalities. For example, when presented with a diagram, readers typically first visually analyze it as a whole to get a general idea of what it describes. From there, they usually focus on the most obvious data or labels first, and then gradually scan the rest of the data and labels to gain a more complete understanding of the information presented.

## 3 Eye Tracking in NLP

### 3.1 Basic Notions

Eye tracking is an important technique for studying eye movements during human reading. Human reading is a complex cognitive activity, and eye movement trajectories can visualize the reading process and are important for understanding how humans acquire and comprehend knowledge. Intuitively, different visual tasks result in varying scan (i.e., eye movement) patterns. Studying these patterns can help us understand the mechanisms of human cognitive processing. Numerous research in neuroscience has established a strong association between eye-tracking data and language comprehension activity in human brains (Henderson and Ferreira, 1993). The eye movement trajectory is typically described as an irregular curve, mainly composed of two alternating eye movement actions: **saccade** and **fixation**. A saccade is when our eyes rapidly jump from one word to the next in the text, representing the shift of attention. Fixation refers to the situation we sometimes focus on a word and remains stationary during the visual task for a period of time. In addition, the scan pattern and speed of the eye movement trajectory can be affected by various factors, such as font size, line spacing, contrast, etc. of the text.

### 3.2 Applications

In cognitive-motivated natural language processing (NLP), several studies have investigated the impact of eye-tracking data on NLP tasks, for example, designing machine learning models for NLP tasks such as part-of-speech tagging (Barrett et al., 2016), term extraction (Yaneva et al., 2017), and syntactic parsing (Kim, 2009). Later, researchers combine eye-tracking data with word embeddings in neural models to improve NLP tasks, including sentiment analysis (Mishra et al., 2017) and named entity recognition (NER) (Hollenstein and Zhang, 2019) or revising neural attention (Barrett et al., 2018; Sood et al., 2020; Takmaz et al., 2020). Recent studies (Bhattacharya et al., 2020; Ren and Xiong, 2021; Ding et al., 2022; Khurana et al., 2023) attempt to align text features and cognitive signals to identify their differences and commonalities.

## 4 Dataset

### 4.1 Data Collection

Our work aims to evaluate how reading order impacts the comprehension of VRDs. To achieve this, we randomly select documents mainly from three available datasets: FUNSD (Jaume et al., 2019), SeaBill (Zhang et al., 2022), and Infographic (Mathew et al., 2022). These datasets are widely used and provide diverse examples of complex structured and graphic documents for our analysis. We reuse all document images in

| | # | WEAK | STRUCTURED | INFOGRAPH | TOTAL |
|---|---|---|---|---|---|
| **Pattern** | | *norm-z* | *local priori* | *cross&visual* | - |
| **Train** | doc | 149 | 160 | 100 | 409 |
| | ent | 7,441 | 10,024 | 12,650 | 30,115 |
| | tok | 22,512 | 16,055 | 24,364 | 62,931 |
| **Test** | doc | 50 | 50 | 30 | 130 |
| | ent | 2,332 | 3,430 | 3,794 | 9,556 |
| | tok | 8,973 | 7,022 | 7,308 | 23,123 |

Table 1: Data statistics in the DOCTRACK dataset categorized by the type of VRDs. We also give the primary scan patterns in each subset.

the FUNSD dataset (Jaume et al., 2019). Given that these documents are less structured than the other datasets and the eye movement tracks primarily to conform to the normal-$Z$ reading order pattern, we rename this sub-dataset as "WEAK." We have selected a number of structured tabular documents from the SEABILL dataset that contains detailed information related to the shipment of goods. This information includes the name of the consignor and consignee, the type and quantity of the shipment, etc. The subset comprises 160 training samples and 50 testing samples, totaling 13,454 semantic entities, denoted as "STRUCTURED." We also select 100 training samples and 30 test samples from the Infographic dataset (Mathew et al., 2022). This subset contains a large number of graphic components, it is more diverse and complicated than the other two subsets, called "INFOGRAPH." Table 1 shows details of the dataset and statistics.

### 4.2 Analysis of Eye Movement Patterns

We observe mainly four types of reading patterns among these documents. Figure 2 illustrates human reading behaviors when reading the documents from the DOCTRACK dataset.

**Normal-Z**. The normal-Z order, also called

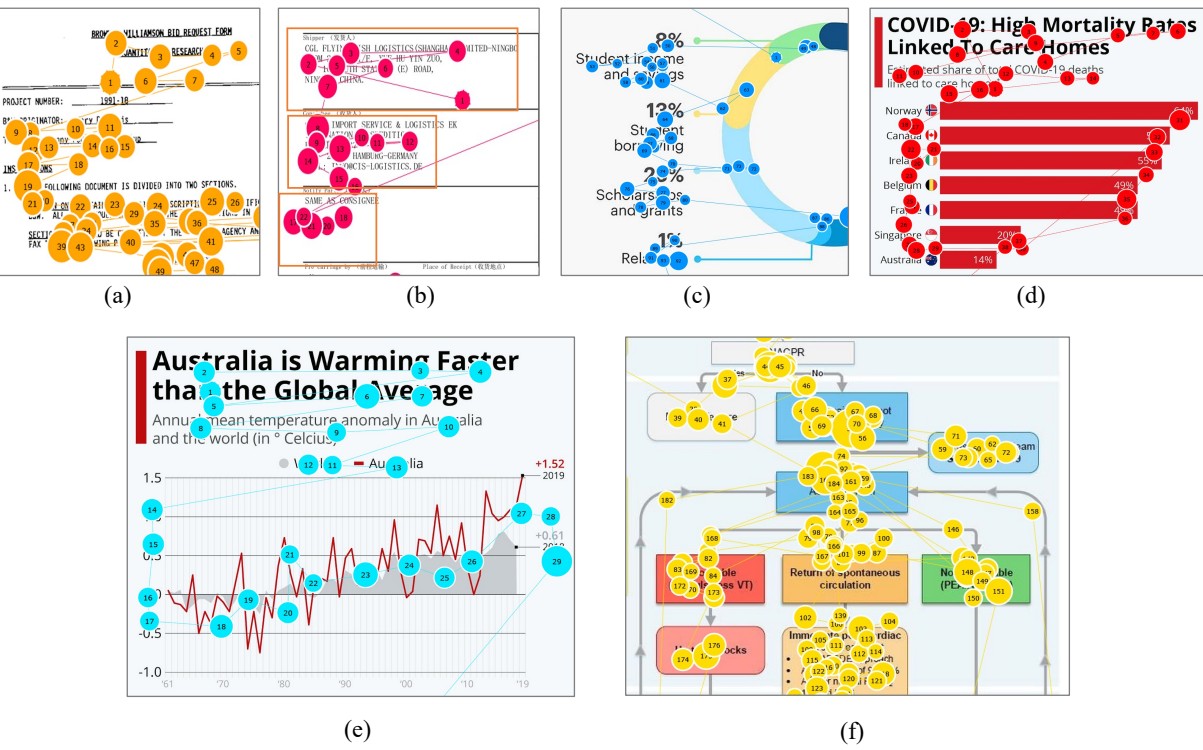

Figure 2: Scatterplots showing eye movement patterns while humans are reading can be organized in various ways: (a) normal-Z order; (b) local priority order; (c, d, e) cross-modal interaction order; (f) visual instruction order. Note that these eye movement data have not yet been aligned with the OCR output. The full version of the images can be found in Appendix A.2.

"zigzag pattern," refers to the typical eye movement pattern that occurs when people read pure text or weakly structured documents. This zigzag pattern is an efficient way for the human visual system to process and understand text. Specifically, our eyes move from left to right along a line of text and then jump back to the beginning of the next line during reading, creating a zigzag scan trajectory of eye movements. Figure 2(a) shows the diagram of eye movement sequences during human reading, which often resembles the shape of the letter "Z".

**Local priority**. Local-priority eye-movement patterns are a common cognitive strategy that humans use when dealing with the hierarchical layout structure of forms or tables. Usually, as shown in Figure 2(b), by prioritizing local information, we focus on the content inside a tabular cell first and then shift our attention to other elements around the cell while looking up in the tabular or a form document. This facilitates quick reading and retrieval of informative text to obtain the needed information.

**Cross-modal interaction**. When people read infographics that contain pies, charts, or other types of data graphs, they typically show interactive radial eye movements. During reading, the eyes follow a radial path between the graph and the associated text. For example, when people read a pie chart, they usually focus first on the pie portion of the chart and then scan along its perimeter to find the corresponding text. This back-and-forth movement results in a cross-modal interaction pattern. The unique shape and layout structure of pie charts contributes to this pattern because the graphical portion often contains the most important information that people attend to first. In contrast, the text label section provides detailed information that people need to scan step by step in order to understand. Therefore, eye tracking presents a radial pattern in these types of infographics.

**Visual instruction**. When reading flowcharts, an eye movement pattern characterized by backtracking often develops, as readers must understand and compare different texts in the boxes. Flowcharts commonly contain a great deal of information and detail presented in a hierarchical, logical manner. Readers typically focus first on the chart's overall structure and theme, then progress gradually to each detail. When encountering text that requires

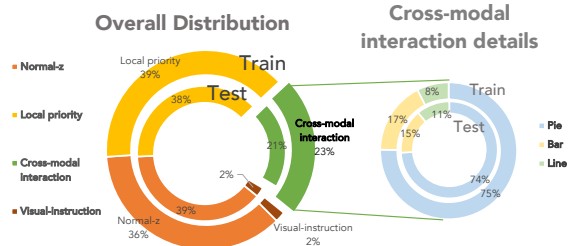

Figure 3: The statistics of four types of reading patterns.

contextual referencing, readers may re-read earlier sections to aid in better understanding the information's meaning. This self-correcting, backtracking eye movement pattern reflects the information processing of the reader and can help with better comprehension and memory retention of the information. Figure 3 shows the statistics of the four types of reading patterns found in DOCTRACK.

### 4.3 Eye-tracking Experiments

To conduct eye-tracking experiments, we randomly divide the dataset into five parts and recruit five participants.[1] All participants are instructed to read the data on an HP 24-inch 1080P display. We record the participants' eye trajectories while they are reading using Tobii TX300 and Tobii Studio. These devices can record the visual movement trajectory of each subject and convert it into digital data. Data extracted from the eye tracker includes fixation points, fixation time, frequency, eye saccade distance, and pupil size. These measurements can help researchers gain insights into participants' internal cognitive processes.

The high sampling frequency of an eye tracker may result in an unsmooth trajectory with recorded gaze points. In addition, gaze points during eye movements may deviate or miss due to peripheral vision (see Appendix A.1). To improve the accuracy of the eye tracker when recording the reading trajectory, we do as follows:

1. In the case of missing gaze points (case 1), if the gaze point hits the periphery of the known OCR bounding box within a certain Euclidean distance, we use the ordinal number of the peripheral gaze point as the reading sequence index of the current bounding box.

2. In the case of missing gaze points (case 2), we use the ordinal number from the surround-

---

[1]All participants are graduate or undergraduate students.

ing adjacent reading sequence or the ordinal number between two gaze points.

3. We delete gaze points that are repeatedly returned to the eye multiple times and keep only the ordinal numbers of the first eye moment.

By adopting the above corrections, we ensure the accuracy of the recorded reading trajectory.

### 4.4 Annotation Agreement

In our final experimental setup, we assign two out of the five participants to label the same subset of data. Subsequently, for each document file, we compare the labeling results from these two participants. We then conduct a voting process among the five participants to select the most appropriate labelingdocument-levelthat aligns with everyone's expectations. This chosen labeling is ultimately considered as the final data.

## 5 Reading Order Integration

### 5.1 Rule-based Heuristic Methods

Multi-modal information extraction methods rely on accurate sequence detection of documents. However, inconsistencies in OCR engines can lead to variances in reading order. To address this issue, Li et al. (2022) introduce position offset threshold to standardize reading order and deal with OCR's instabilities. The text boxes are sorted from top to bottom, and if the distance between two boxes in the $Y$ direction is smaller than the threshold, then their order is determined based on their $X$ direction order. Besides, Gu et al. (2022) sort the bounding box according to the two coordinates of the $Y$-axis, and then performs the $X$-axis search and the right-down search with $Y + X$ combined consideration. By changing the order of token input into the model, both approaches achieve good results. The proposed rule-based sorting approaches conform to the basic cognition of humans when reading, resulting in accurate information extraction. Therefore, leveraging the reading order generated by rule-based sorting approaches can significantly improve the accuracy of multimodal information extraction.

### 5.2 Modal-based Preordering Methods

#### 5.2.1 Proposed Pipeline

To follow human reading behaviors, we use a process called "preordering" (reordering-as-

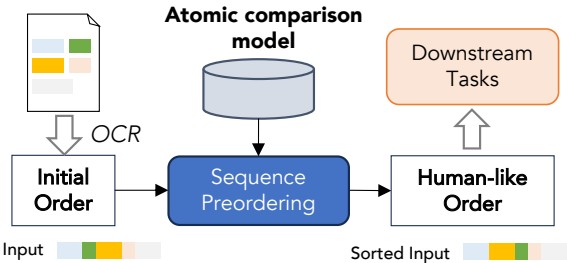

Figure 4: The proposed preordering pipeline for utilizing human-like reading orders in VRD understanding tasks.

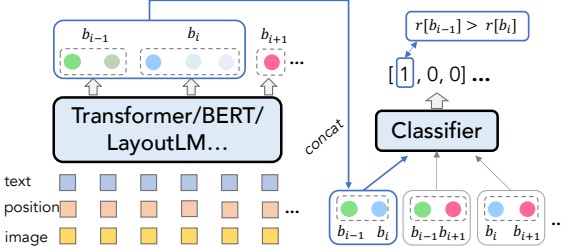

Figure 5: Atomic reading order comparison model for neighboring bounding boxes. The symbol ">" indicates that the former bounding box might be read by humans after the latter.

preprocessing), a term borrowed from the domain of statistical machine translation (Xia and McCord, 2004; Collins et al., 2005; Neubig et al., 2012; Nakagawa, 2015). This process involved reorganizing the inputs in the order they would be read by a human. By this, we make it easier for the reader to understand the procedure of aligning the multimodal sequential input features with human reading order and thus enabling us to evaluate the impact of reading order on the VRD understanding tasks.

### 5.2.2 Atomic Comparison Models

We obtain the basic features of different modalities of documents through different encoders and imitate human reading order according to these features. Therefore, we propose four different models for reading order generation. Each model uses information from single or multiple modalities simultaneously: **Box**, **Text**, **Text+Box**, and **Text+Box+Image**.

Each model takes into account different factors that influence how humans prioritize and read elements in VRDs, including the position of the element, the text within the element, and the visual region associated with it. By utilizing these models, we can more accurately evaluate the impact of reading order on human comprehension of such documents.

$$p = f(b_i : b_j) = \begin{cases} 0, & \text{if } r[b_i] < r[b_j] \\ 1, & \text{if } r[b_i] > r[b_j] \end{cases} \quad (1)$$

where $b_i$ and $b_j$ denote the $i$-th and $j$-th bounding boxes, respectively, $r[b_i]$ refers to the index ID of the bounding box $b_i$ in the predicted reading sequence, and $r[b_i] < r[b_j]$ indicates that $b_i$ appears before $b_j$ in the reading order sequence.

**Box**. In this model, we use only two-dimensional positions to learn the order of each bounding box. Specifically, at first, the given bounding box coordinates $(x_{\text{up}}, y_{\text{up}}, x_{\text{down}}, y_{\text{down}})$ (representing the top-left and bottom-right coordinates) generated by the OCR tool are used to compute the centroid coordinates $(x_i, y_i)$ of each bounding box. We combine all the bounding box centroid coordinates in pairs, then feed the centroid coordinates of the two combined bounding boxes directly into a multi-layer Transformer network. This enables us to predict the spatial relationship between the two bounding boxes, i.e., which one should come before or after in the human reading order.

$$b_i : b_j = \text{Transformer}(x_i, y_i, x_j, y_j) \quad (2)$$

**Text**. We use BERT to encode the texts, and since each bounding box contains one or more tokens, we take the latent representation at the first token position as the embedding for a bounding box.

$$b_i : b_j = \text{BERT}(\mathbf{t}_i, \mathbf{t}_j) \quad (3)$$

**Text+Box**. To jointly encode the text and 2D-positions inputs, we use LayoutLM. Similar to the operation in the **Text** section, we take the first latent representation as the input to the classifier. The final model can be formulated as follows:

$$b_i : b_j = \text{LayoutLM}(\mathbf{t}_i, \mathbf{t}_j; x_i, y_i, x_j, y_j) \quad (4)$$

**Text+Box+Image**. We use LayoutLMv2 to joint encode the text, image and 2D-position within the bounding boxes as follows:

$$b_i : b_j = \text{LayoutLMv2}(\mathbf{t}_i, \mathbf{t}_j; x_i, y_i, x_j, y_j; \mathbf{I}_i, \mathbf{I}_j) \quad (5)$$

where we consider the ROIs of document images, denoted as $\mathbf{I}_i$ and $\mathbf{I}_j$. We take the first latent encoding to represent each bounding box.

| Modality | WEAK | | STRUCTURED | | INFOGRAPH | | OVERALL | |
|---|---|---|---|---|---|---|---|---|
| | $\tau \uparrow$ | $\rho \uparrow$ | $\tau \uparrow$ | $\rho \uparrow$ | $\tau \uparrow$ | $\rho \uparrow$ | $\tau \uparrow$ | $\rho \uparrow$ |
| BOX | 0.4521 | 0.4731 | 0.7017 | 0.7411 | 0.6761 | 0.7413 | 0.5992 | 0.6366 |
| TEXT | 0.5369 | 0.5740 | 0.8965 | 0.9717 | 0.9046 | 0.9171 | 0.7589 | 0.8052 |
| TEXT+BOX | 0.5316 | 0.5749 | **0.9269** | **0.9766** | **0.9717** | **0.9977** | 0.7837 | 0.8256 |
| TEXT+BOX+IMAGE | **0.6293** | **0.6893** | 0.8930 | 0.9690 | 0.9575 | 0.9954 | **0.8052** | **0.8665** |
| MISSING RATE | 38.16% | | 12.98% | | 9.55% | | - | |

Table 2: Correlation coefficient scores of Kendall's tau $\tau$ and Spearman's rank $\rho$ between resorted input sequences generated by using different preordering methods and golden human eye movement orders. **Bold** indicates the best and underline indicates the second best.

---

**Algorithm 1** Model-based Sequenece Preordering

**Input:** the serialized input sequence $[b_1, \ldots, b_l]$ that contains multimodal features ;
**Output:** the sorted input sequence $\mathbf{r}$;

$\mathbf{r} \leftarrow [b_1, \ldots, b_l]$;
**for** $i \leftarrow 0$ **to** $l - 2$ **do**
    **for** $j \leftarrow 0$ **to** $l - i - 2$ **do**
        $p \leftarrow f(\mathbf{r}_j : \mathbf{r}_{j+1})$; ▷ Calling the atomic comparison model to determine the precedence order.
        **if** $p < 0.5$ **then**
            swap $\mathbf{r}_j$ with $\mathbf{r}_{j+1}$; ▷ Consistent with Bubble sort.
**return** $\mathbf{r}$; ▷ New sorted sequence.

---

### 5.2.3 Sequence Preordering Algorithm

We test four atomic comparison models within the model-based sequence preordering algorithm as a before-and-after judgment. The preordering algorithm takes as input the atomic comparison models outputs (0/1 sequence) to construct an adjacent matrix. Thus, the preordering algorithm is a variant of the commonly used Bubble Sort algorithm, which outputs the sorted input sequence by rearranging the bounding box positions in the sequence. Refer to Algorithm 1 for more details.

## 6 Experiments and Analysis

To gain a deeper understanding of the differences between human reading order and machine processing order, we conduct both intrinsic and extrinsic evaluations using the DOCTRACK dataset. By comparing different modal fusions, the intrinsic evaluation allows us to assess the quality of the reading order generation model, which in turn helps us to understand the extent to which information from each modality influences the reading order built by humans. In order to determine ex-

actly what reading order is needed for a machine document intelligence model, i.e., what input order enhances machine document comprehension, we make use of extrinsic evaluation to assess the quality of temporal order on human-like reading order generation.

### 6.1 Intrinsic Evaluation

Specifically, we compare machine-generated input orders with ground truths, i.e., human reading orders, to evaluate the model's performance in the intermediate task of reading order generation. We measure the correlation between the machine-generated input order and the reading order of human experts by calculating Spearman's rank and Kendall's tau scores to assess the accuracy of the machine-generated reading order and the interaction between different modalities and different types of documents.

Table 2 shows the sequence correlation evaluation results of four different models on three datasets. Among them, the average result of the model based on Text+Box+Image is the best, and the result based on the bounding box is the worst, which also verifies multimodal features have an important impact on document ranking. The results in the above table show that humans read VRDs with weak table structure, and humans still use visual features such as font background to help model the temporal sequence, and visual features are less important in the case of table structure, and basic text and position are enough. However, infographics with more images and visual features require more powerful models that can capture text, location, and visual features. For more details see Appendix A.3.

| | PREORDER | MODALITY | | | TYPE | WEAK | | | STRUCTURED | | | INFOGRAPH |
|---|---|---|---|---|---|---|---|---|---|---|---|---|
| | | Text | Pos | Img | R/H/M | P (%)↑ | R (%)↑ | F1 (%)↑ | P (%)↑ | R (%)↑ | F1 (%)↑ | ANLS (%)↑ |
| **BERT** | EYE | ✓ | | | H | 57.75 | 60.23 | 58.70 | 57.63 | 59.13 | 58.37 | 4.01 |
| | EYE++ | ✓ | ✓ | | H | 60.52 | 60.77 | 60.47 | 60.75 | 60.83 | 60.79 | 4.65 |
| | DEFAULT-OCR | ✓ | | | R | 56.69 | 62.11 | 60.33 | 58.99 | 60.01 | 59.51 | 3.82 |
| | Z-ORDER | ✓ | ✓ | | R | **64.09** | **65.28** | **64.66** | **63.44** | **62.78** | **63.11** | **5.88** |
| | XYLAYOUT | ✓ | ✓ | | R | 60.16 | 60.84 | 60.19 | 59.24 | 60.08 | 59.65 | 3.71 |
| | MODEL-B | ✓ | ✓ | | M | 60.19 | 61.98 | 60.92 | 59.01 | 60.98 | 59.98 | 2.94 |
| | MODEL-T | ✓ | | | M | 62.80 | 63.16 | 62.87 | 61.04 | 60.22 | 60.62 | 2.99 |
| | MODEL-T+B | ✓ | ✓ | | M | 61.30 | 62.43 | 61.80 | 60.52 | 60.77 | 60.47 | 3.14 |
| | MODEL-T+B+I | ✓ | ✓ | ✓ | M | 62.74 | 64.51 | 63.45 | 62.43 | 62.17 | 62.80 | 3.21 |
| **LayoutLMv2** | EYE | ✓ | | | H | 82.13 | 85.11 | 83.58 | 77.84 | 74.14 | 75.94 | 14.43 |
| | EYE++ | ✓ | ✓ | | H | 87.38 | 83.82 | 85.41 | 77.44 | 74.39 | 75.88 | 15.69 |
| | DEFAULT-OCR | ✓ | | | R | 86.94 | 80.95 | 83.44 | 78.56 | 73.02 | 75.69 | 12.50 |
| | Z-ORDER | ✓ | ✓ | | R | **88.00** | 84.46 | 86.06 | 78.05 | 74.77 | 76.37 | **18.09** |
| | XYLAYOUT | ✓ | ✓ | | R | 84.01 | 83.12 | 83.55 | 75.01 | **78.41** | 76.61 | 12.38 |
| | MODEL-B | ✓ | ✓ | | M | 87.23 | 85.16 | 86.13 | 78.45 | 73.01 | 75.63 | 12.98 |
| | MODEL-T | ✓ | | | M | 85.35 | 82.40 | 83.77 | 77.64 | 77.14 | 77.39 | 12.67 |
| | MODEL-T+B | ✓ | ✓ | | M | 86.76 | 86.61 | 86.57 | **80.24** | 74.84 | 77.45 | 14.70 |
| | MODEL-T+B+I | ✓ | ✓ | ✓ | M | 87.00 | **87.01** | **86.98** | 79.74 | 75.58 | **77.60** | 14.70 |
| **LayoutLMv3** | EYE | ✓ | | | H | 91.46 | 90.49 | 90.97 | 68.29 | 63.82 | 65.98 | 17.91 |
| | EYE++ | ✓ | ✓ | | H | 91.47 | 91.19 | 91.33 | 69.22 | 65.57 | 67.35 | 18.64 |
| | DEFAULT-OCR | ✓ | | | R | 90.96 | 92.00 | 91.48 | 72.96 | 66.71 | 69.70 | 18.22 |
| | Z-ORDER | ✓ | ✓ | | R | **94.63** | 92.85 | **93.73** | **77.27** | 68.24 | 72.47 | **20.58** |
| | XYLAYOUT | ✓ | ✓ | | R | 90.10 | 89.69 | 89.90 | 71.05 | 66.73 | 68.82 | 17.63 |
| | MODEL-B | ✓ | ✓ | | M | 92.37 | 91.95 | 92.16 | 75.81 | 70.00 | 72.79 | 17.76 |
| | MODEL-T | ✓ | | | M | 91.86 | 93.09 | 92.47 | 74.01 | 67.56 | 70.64 | 17.52 |
| | MODEL-T+B | ✓ | ✓ | | M | 92.39 | 92.85 | 92.62 | 77.17 | **70.98** | **73.94** | 18.01 |
| | MODEL-T+B+I | ✓ | ✓ | ✓ | M | 93.98 | **93.14** | 93.56 | 74.17 | 68.97 | 71.47 | 18.24 |

Table 3: Extrinsic evaluation on the DOCTRACK dataset. DEFAULT-OCR represents the original order, EYE and EYE++ represent the original eye movement order and the smoothed eye movement order, Z-ORDER (Yu et al., 2023b) and XYLAYOUT (Gu et al., 2022) are two orders generated by expert experience, MODEL-B, MODEL-T, MODEL-T+B and MODEL-T+B+I represent atomic comparison models such as Box, Text, Text+Box, and Text+Box+Image, respectively. R/H/M refers to the order generated by rules, humans, and models.

## 6.2 Extrinsic Evaluation

**Semantic entity recognition**. The purpose of this study is to analyze the impact of reading order on the VRD understanding using the Semantic Entity Recognition (SER) task for the WEAK and STRUCTURED subset of documents. Table 3 shows the results. We find that the impact of reading order on the SER task varies depending on the document AI model used. When using BERT, the simple Z-order works best, and the effect of each order is better than the effect of the original order. In the multimodal LayoutLMv2 and LayoutLMv3 models, multimodal ordering works best, slightly better than Z-order. These results suggest that human reading order and machine ordering order have a strong influence on the SER task, and different models have different degrees of sensitivity to these orders.

**DQA**. Document Question Answering (DQA) is a challenging task in VRD understanding that requires machines to understand both the visual and textual content of a document image and to answer questions about it. Table 3 lists the Average Normalized Levenshtein Similarity (ANLS) scores on the INFORGRAPH subset of text-only baseline BERT, layout-aware multimodal baselines LayoutLMv2 and LayoutLMv3. We observe that LayoutLMv2 and LayoutLMv3 models outperform text-only baselines (BERT) by a large margin. While integrating human reading order enhances the state-of-the-art document AI model in downstream tasks, it does not always outperform the human-like reading order generated using a rule-based approach. This suggests that true human reading order may not be necessary to enhance existing machine document AI models.

There are several possible reasons for this. First, most of the datasets used by existing document AI models are sorted by simple rules and therefore are better suited for the orders generated by using simple rules such as $Z$-pattern. Additionally, individual human reading orders may be very noisy unless a large human eye-movement dataset is constructed by collecting a significant amount of human eye-movement data.

## 7 Conclusion

We investigate the impact of human reading order on Document AI models for VRD understanding tasks. We propose different methods to generate human-like reading orders, along with a practical preordering pipeline that can leverage the generated reading orders. Our observations suggest that true human reading order may not always be suitable for reading VRDs. The dataset we construct can help in designing better document AI models and human reading robots in the future.

## Limitations

In this work, we focus on the impact of human eye-tracking order and machine reading ordering for VRD understanding. Due to the complexity of eye movement characteristics, when the participants were doing eye movement experiments, they were required to ignore the eye movements information, such as the fixation time of each fixation point, back gaze, and the number of fixations. Therefore, our next step will be to explore the impact of more eye movement gaze information on the independent understanding of VRDs. In addition, due to the high annotation cost, the annotation has not been done by multiple annotators. Therefore, the inner-agreement rate is not available for the current dataset.

## Acknowledgements

This work was supported by National Natural Science Foundation of China (Young Program: 62306173, General Program: 62176153), JSPS KAKENHI Program (JP23H03454), Shanghai Sailing Program (21YF1413900), Shanghai Pujiang Program (21PJ1406800), Shanghai Municipal Science and Technology Major Project (2021SHZDZX0102), the Alibaba-AIR Program (22088682), and the Tencent AI Lab Fund (RBFR2023012).

## Ethics Statement

This study has received institutional ethics approval and complies with the Declaration of Helsinki and subsequent revisions. Informed consent was obtained from all participants before the study began, and they were informed that they had the right to withdraw from the study at any time. Personal identities are removed from the data to ensure anonymity. The participants have also approved eye-tracking data release for research purposes.

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

# A  Appendix

## A.1  Visualization of Human Reading Order

Figure 6 shows the missing gaze points during human reading and after smoothing.

## A.2  Human Reading Patterns

Figure 7 shows four patterns of human reading orders.

## A.3  Missing Gaze Points Visualization

Figure 8 shows the sequence of model generation and the sequence of human eye gaze.

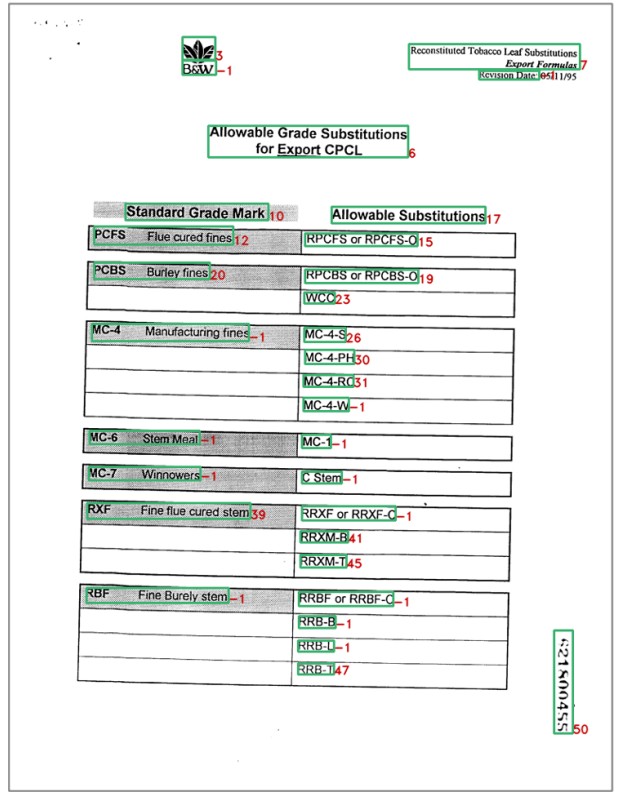

(a) missing gaze points

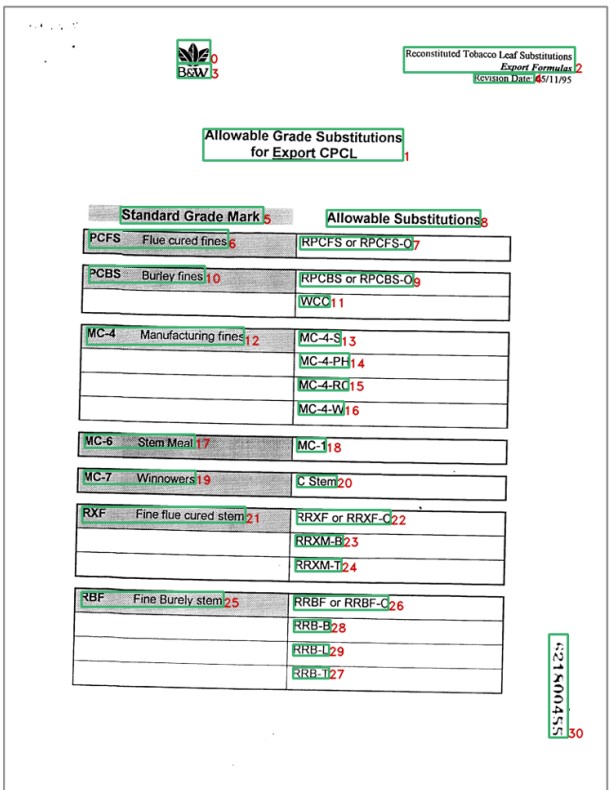

(b) fixed data visualization

Figure 6: Example of a document image with missing gaze points (-1) and fixed data in the WEAK subset.

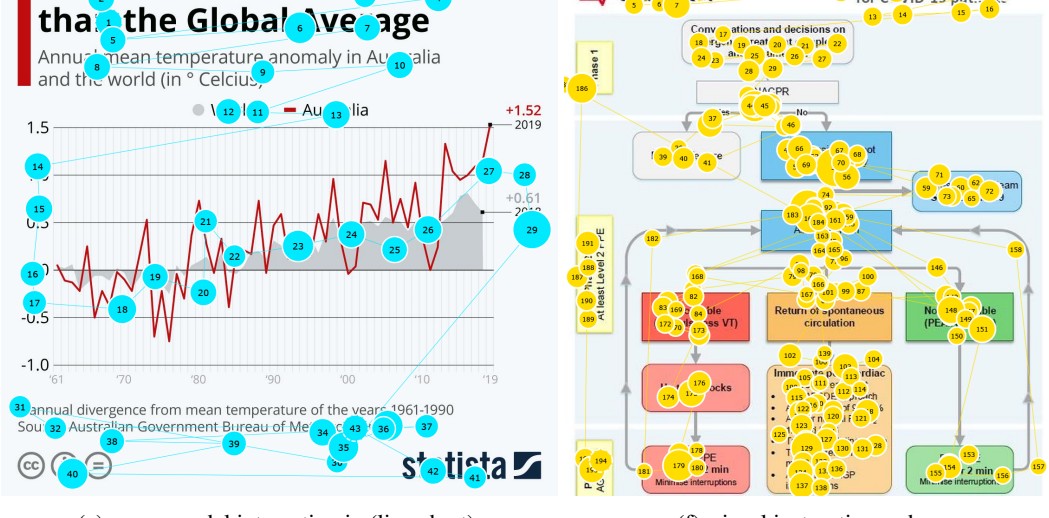

(a) normal-Z order

(b) local priority order.

(c) cross-modal interaction (pie chart).

(d) cross-modal interaction (bar chart).

(e) cross-modal interaction in (line chart).

(f) visual instruction order.

Figure 7: A scatterplot showing four patterns of human eye movement while reading.

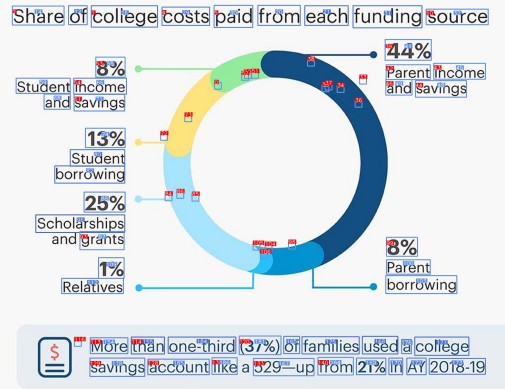

(a) Sequence diagram of human eye movements while reading

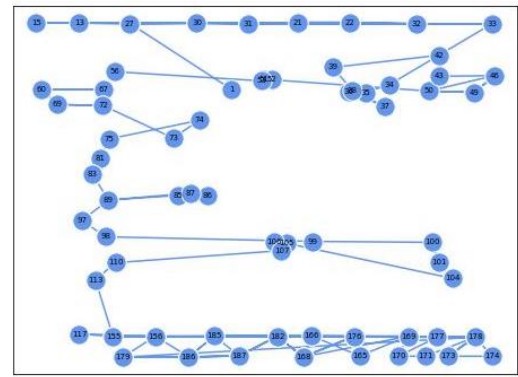

(b) Scatterplot of human eye movement when reading.

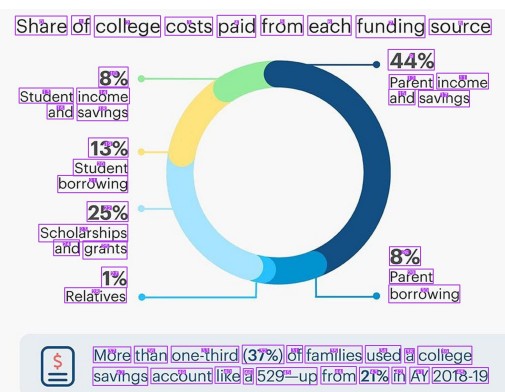

(c) Sequence diagram of reading order generated by MODEL-B+T+I.

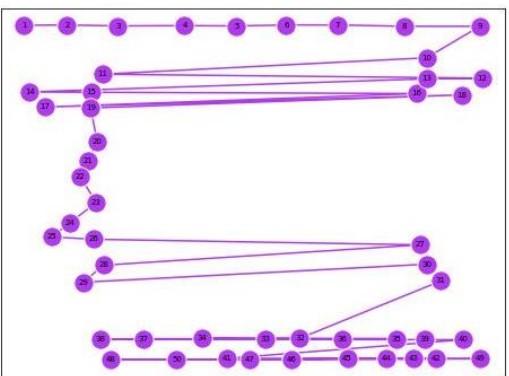

(d) Scatterplot of reading order generated by MODEL-B+T+I.

Figure 8: Case comparison of human reading order and the reading order generated by MODEL-B+T+I.