# OpenReview forum: "DocTrack: A Visually-Rich Document Dataset Really Aligned with Human Eye Movement for Machine Reading"
_EMNLP/2023/Conference — EMNLP 2023 Findings_

### Official Review · Reviewer_sdSf · 2023-08-02

**Soundness:** 3

**Excitement:**

4: Strong: This paper deepens the understanding of some phenomenon or lowers the barriers to an existing research direction.

**Paper Topic And Main Contributions:**

The research paper addresses a key challenge in the field of AI - understanding Visually-Rich Documents (VRDs). VRDs are documents with substantial visual components such as tables, graphs, diagrams etc., which are increasingly common due to advancements in information technology. Document AI models are employed to comprehend these VRDs, but their understanding is often limited by the inability to accurately follow human-like reading order and their struggle to comprehend the complex interplay of text, visuals, and layout. To address these limitations, the authors introduce DOC TRACK, a unique VRD dataset that incorporates human eye-movement information gathered using eye-tracking technology. The dataset aims to explore how human reading patterns might influence document comprehension tasks, presenting a comparison between the human reading order and how AI models read and comprehend documents.

The paper makes three main contributions:

1. It presents DOC TRACK, the first-of-its-kind dataset, aligned with human eye movement data. This benchmark dataset is expected to facilitate research on reading order generation in VRDs, thereby enhancing our understanding of how humans and machines differ in their approach to reading and interpreting VRDs.

2. The authors explore various methods to generate human-like reading orders and suggest a practical preordering pipeline. This pipeline integrates human reading order into the AI model workflow, which has the potential to improve the models' document understanding capabilities.

3. Lastly, the authors conduct a detailed evaluation of the impact of human-like reading order on AI models' performance in document understanding tasks. These evaluations help determine the efficacy of human reading order when used by AI in understanding VRDs, which may lead to more refined AI models in the future. Their findings indicate that despite improvements in Document AI models, they still have substantial progress to make before they can accurately and effectively understand VRDs like humans do.


**Reasons To Accept:**





1. The paper introduces DOC TRACK, a first-of-its-kind dataset that aligns VRDs with human eye movement data. This unique benchmark can pave the way for new research in the field of VRD reading order generation.
2. The proposed preordering pipeline, which incorporates human-like reading orders into AI models, is an interesting and practical solution to improve VRD understanding.
3. The authors conduct both intrinsic and extrinsic evaluations, providing a thorough analysis of the impact of human-like reading orders on VRD comprehension tasks.




**Reasons To Reject:**



1. According to Table 1, the dataset does not have a valid/dev set, which may add some difficulties in tuning the system.
2. It would be better to open-source the baseline systems to further facillitate the community.




**Reproducibility:**

3: Could reproduce the results with some difficulty. The settings of parameters are underspecified or subjectively determined; the training/evaluation data are not widely available.

**Reviewer Confidence:**

2: Willing to defend my evaluation, but it is fairly likely that I missed some details, didn't understand some central points, or can't be sure about the novelty of the work.

---

> ### Author Rebuttal · Authors · 2023-08-27
>
> ***Response-Q1***: It's great to see that the reviewer raise the issue of considering the importance of validation and development sets. Taking into account the limited number of annotations available, currently, there is no designated validation or development set.
>
> Nevertheless, validation sets in machine learning can be employed for cross-validation, such as by allocating a portion of the training set to serve as a validation set.
>
> As for additional dev/valid set, we appreciate this valuable suggestion. We will contemplate further annotating data to establish an additional validation set and release it in the not-too-distant future.
>
> ***Response-Q2***:  We have committed to openly sharing all preordering models, data sets, and naturally, baseline systems.

---

### Official Review · Reviewer_hWn3 · 2023-08-04

**Typos Grammar Style And Presentation Improvements:** Gaza -> Gaze ? in the appendix.
**Soundness:** 3

**Excitement:**

4: Strong: This paper deepens the understanding of some phenomenon or lowers the barriers to an existing research direction.

**Missing References:**

N/A

**Paper Topic And Main Contributions:**

This work towards document reading aims to align the eye movement during a human's actual reading with existing resources and models, revealing the findings that natural human eye movement modeling is beyond the current VRD models' abilities to make the process more human-like, where these models still depend highly on rules and other assumptions.

This work constructs a real human eye movement dataset for further research of document AI systems.

**Questions For The Authors:**

1. Can WEAK conforming to the normal-Z pattern be regarded as the easiest part of the dataset and InfoGraphics the most difficult one requiring more complicated modeling? If so, models can better fit into WEAK preordering when doing intrinsic evaluations. However, from Tab. 2, correlation coefficient scores on STRUCTURED and INFOGRAPHICS (especially INFOGRAPHICS) are much higher than WEAK. Why would this happen? Is there any further explanation?
2. For extrinsic evaluations, why do EYE and EYE++ show such different behaviors to Model-T and Model-T+B on STRUCTURED and INFOGRAPH as Model-T(T+B) has achieved extremely high correlations with labels in the preordering?
3. How often does the missing gaze point be fixed by the strategies in 4.3? How to measure and guarantee the accuracy of the recording with these strategies? Some measurements on a small subset can be more convincing. Is there any possibility that the dataset can be affected by these strategies so that WEAK is harder to fit?

**Reasons To Accept:**

* A new resource for real human eye movements aligned in the document reading modeling.
* A good point of view to find the discrepancy between real human behavior and the current models' modeling based on rules.
* A work with good writing can provide a comfortable reading experience.

**Reasons To Reject:**

* The appeared results of some experiments contradict the dataset's setup and assumptions. No further analyses or experiments are adapted to explain them. See Questions For The Authors.
* More details about the dataset construction are required.

**Reproducibility:**

4: Could mostly reproduce the results, but there may be some variation because of sample variance or minor variations in their interpretation of the protocol or method.

**Reviewer Confidence:**

3: Pretty sure, but there's a chance I missed something. Although I have a good feel for this area in general, I did not carefully check the paper's details, e.g., the math, experimental design, or novelty.

---

> ### Author Rebuttal · Authors · 2023-08-27
>
> At the suggested by the reviewers, we conducted further experiments. We discovered that the rate of missing gaze points significantly impacts the results of both intrinsic evaluation and extrinsic experiments, and may be the primary cause of different behaviors.
>
> ***Response-Q1: The models did not yield the expected performance on the WEAK subset preordering during intrinsic evaluations.***
>
> This issue indeed requires clarification.
> some explanation can be found in ***Response-Q3***.
>
>
> ***Response-Q2:EYE/EYE++ different behaviors to Model-T/Model-T+B***:
>
> We think the explanation for this question can also be found in ***Response-Q3***.
>
> ***Response-Q3:missing-gaze-point-rate***
>
> We greatly appreciate this question, and we must admit that we had not observed this phenomenon previously!
>
> For missing gaze points (in **eye**), we use -1 as a placeholder. Given a document order of [1, -1, 2, 5, -1, 6], we determine the current order of missing points based on the points before and after them. We then insert the missing points into the gaze point sequence behind them according to their coordinates, resulting in [1, 2, 2, 5, 5, 6] (in **eye++**). This approach makes the document order easier to fit rather than more difficult.
>
> Following the reviewer's advice, we calculated the ratio of missing gaze points for each specific subset by determining the proportion of -1 points relative to the total, see result below.
>
> |                  |  WEAK  | STRUCTURED | INFOGRAPH |
> | ---------------- | :----: | :--------: | :-------: |
> | missing rate | **38.16%** |   12.98%   |   9.55%   |
>
>
> After an extensive discussion, we endeavored to provide explanations to the best of our knowledge as follows:
>
> 1) Due to the involuntary fluttering of human eyes while reading, the gaze point may deviate from the intended text area.
>
> 2) The text segments in the document image are dispersed across the page, resulting in numerous missing points and a high rate of omissions.
>
> 3) The document layout within the WEAK subset appears simplistic and seemingly arbitrary, leading to a significant discrepancy between the order comprehended by machines (z) and humans (arbitrary).
>
> 4) We also inquired with specific participants, who reported that working on the WEAK dataset was not particularly laborious compared to other two subsets; however, they experienced a heightened level of mental concentration when handling the other two subsets.
>
> Therefore, we believe that the combination of all the aforementioned factors has led to the emergence of questions 1, 2, and 3.

---

### Official Review · Reviewer_2oER · 2023-08-08

**Typos Grammar Style And Presentation Improvements:** N/A
**Soundness:** 4

**Excitement:**

3: Ambivalent: It has merits (e.g., it reports state-of-the-art results, the idea is nice), but there are key weaknesses (e.g., it describes incremental work), and it can significantly benefit from another round of revision. However, I won't object to accepting it if my co-reviewers champion it.

**Missing References:**

N/A

**Paper Topic And Main Contributions:**

This paper proposes to improve the visually-rich document understanding task through the reading order. The new ordering comes from the eye tracking techniques and this new order makes it possible for the language model to understand the documents in a similar way as real human. Experiments show that the current layout-aware LMs can detect the new reading order and the performance is improved with the help of this new ordering, compared with other rule-based ordering techniques.

**Questions For The Authors:**

1. What is the OCR engine used in the paper?
2. Is the reading order phrase-level? This means the eye-tracking technique is used to sort different phrases instead of words. If so, there is a question, how can we guarantee the quality of the word grouping in the OCR engine.
3. Why can the human's reading order benefit the machine to understand the document. As in figure 1, the key value pair of author field is actually reversed.

**Reasons To Accept:**

1 . The reading order is a big issue for visually-rich document understanding. A wrong reading order could make it impossible for simple LMs to get the right results. In this paper, the eye tracking is a strategy to get the ordering information.
2. The experiments show that the new ordering information surpass existing rule-based ordering techniques.

**Reasons To Reject:**

A few important points are missing in the paper:
1. During the annotation, did the authors compare the agreement between different annotators?
2. It is also great to know more about the metric method in the table 2.
3. The OCR is part of the preprocessing and the paper didn't mention which OCR engine they are using. Some OCR engines already integrate the reading order in their algorithms so the reading order result can be much better than what is shown in figure 1.
4. It is not intuitive to ask machines to understand the document in the same order as humans. As shown in figure 1, in the human's reading order, the author's key and values are reversed (label 6 and 7)

**Reproducibility:**

4: Could mostly reproduce the results, but there may be some variation because of sample variance or minor variations in their interpretation of the protocol or method.

**Reviewer Confidence:**

4: Quite sure. I tried to check the important points carefully. It's unlikely, though conceivable, that I missed something that should affect my ratings.

---

> ### Author Rebuttal · Authors · 2023-08-27
>
> Thank you for your valuable comments!
>
> ***Response-Q1: Annotation Agreement***
>
> In our experimental setup, we assign two out of the five participants to label the same subset of data. Subsequently, for each document file, we compare the labeling results from these two participants. We then conduct a voting process among the five participants to select the most appropriate labeling（document-level）that aligns with everyone's expectations. This chosen labeling is ultimately considered as the final data.
>
> ***Response-Q2: metrics***
>
> We are using Kendall's Tau and Spearman's rank correlation coefficient. We will add the details into the Appendix.
>
> - Kendall's Tau is a statistical method used to measure the correlation between two variables. It is mainly used to assess the degree of correlation between the ordering or rank of two variables.The value of Kendall's Tau ranges from -1 to 1, where:1 indicates perfect positive correlation,0 indicates irrelevant,-1 indicates perfect negative correlation.The formula for the Kendall's Tau rank correlation coefficient is as follows:
>
> $$
> \tau=s^{\frac{1}{2}}\sqrt{\frac{(n_1-1)(n_1-2)}{(n_1-1)n_2+(n_2-1)(n_2-2)}}
> $$
>
> where $\tau$ is the Kendall correlation coefficient, $s$ is the sample standard deviation, and $n_1$ and $n_2$ are the frequencies of the two variables respectively,i.e. the number of occurrences of each value taken.
>
> - Spearman's rank correlation coefficient is a nonparametric statistical measure of the degree of association between two sets of rank data. It calculates the correlation based on the rank of the original data (rather than the original data values). The formula for the Spearman rank correlation coefficient is as follows:
>
> $$
> \rho=1-6\frac{d^2_1}{n}-6\frac{d^2_2}{n}+6\frac{d_1d_2}{n}
> $$
>
> where $\rho$ is the Spearman correlation  coefficient,$d_1$ and $d_2$ are the rank data for each of the two variables, and $n$ is the sample size.
>
> ***Response-Q2: OCR engine used***
>
> We use Tesseract and Google Vision OCR engines.
>
> ***Response-Q4: phrase-level order***
>
> The reading order in our study is determined at the segment level, which can alternatively be referred to as the phrase-level. As we only modify the segment-level order of the output generated by the OCR engine, there is no need to be concerned about word grouping, as it has already been addressed. Consequently, our sorted results are guaranteed to maintain consistency with the OCR outcomes.
>
> ***Response-Q5: sometimes reading order reversed***
>
> The reviewer raises an important point. When it comes to human reading, it is indeed inevitable that occasional reversals may occur; however, these instances are typically isolated, and the overall reading process remains orderly. On the other hand, for machines, the optimal reading order for humans may not necessarily be the most efficient or effective approach. It is indeed intriguing to consider that the reading order of documents encountered during LMM pre-training tends to follow a norm order resembling a Z-shape. This particular order appears to be more compatible and user-friendly for these LMMs, enhancing their overall performance.

---

### Official Review · Reviewer_ku22 · 2023-08-11

**Soundness:** 3

**Excitement:**

3: Ambivalent: It has merits (e.g., it reports state-of-the-art results, the idea is nice), but there are key weaknesses (e.g., it describes incremental work), and it can significantly benefit from another round of revision. However, I won't object to accepting it if my co-reviewers champion it.

**Paper Topic And Main Contributions:**

This paper presents a dataset for multimodal reading comprehension aligned with eye-tracking data. Specifically, the authors investigated
the connection between human reading order and how it impacts the comprehension of visually-rich documents (VRD).

**Reasons To Accept:**

- An eye tracking aligned dataset for VRD to understand how humans and machines attend to reading

**Reasons To Reject:**

The sample size of human participants is five, a small number

**Reproducibility:**

5: Could easily reproduce the results.

**Reviewer Confidence:**

4: Quite sure. I tried to check the important points carefully. It's unlikely, though conceivable, that I missed something that should affect my ratings.

**Typos Grammar Style And Presentation Improvements:**

Typo
Buddle --> Bubble sort on page 6, Section 5.2.3

Section 6.2
VQA: A more appropriate subheading will be visual document question understanding [1] or document question answering [2]

References
1. https://rrc.cvc.uab.es/?ch=17&com=introduction
2. https://huggingface.co/tasks/document-question-answering

---

> ### Author Rebuttal · Authors · 2023-08-27
>
> We are delighted that the reviewer has accurately  understood our motivation and highlighted the significance and necessity of creating a dataset for human reading order on visually-rich documents, utilizing eye-tracking techniques.
>
> ***Response-Q1 "The sample size ... is small".***:
>
> - **The experimental process is time-consuming and laborious**. Each time the experiment is conducted, we need to immobilize the participant's body and perform the experiment in a closed environment. The long time labeling process can be challenging for regular individuals to tolerate, resulting in many volunteers opting to quit after only a few labeling sessions. Consequently, the authors had to undertake the data labeling themselves.
> - Unlike other NLP task labeling, **the variance in labeling results among different subjects in the eye-tracking experiment is substantial**, making it nearly impossible to achieve consistent results across two subjects. The labeling outcomes vary significantly among individuals, and having too many participants can adversely impact the labeling results, making it difficult for the machine to learn effectively.
> - **Despite its small size, we appreciate your understanding of the importance of our dataset.** We would like to clarify that our work concentrates on human reading order and its potential positive impact on developing machine document understanding models. We hope that reviewers will give due consideration to our efforts, as the dataset, though small, serves as a crucial foundation for the entire project. The dataset's significance lies not only in its detailed construction process but also in the numerous valuable attempts and intriguing findings derived from it. We believe that these aspects make our paper particularly meaningful to the community and appealing to readers.
>
>
> ***Response-Q2: Typos***
>
> We will correct them.
>
> ***Response-Q3: VQA->DQA***
>
> Agree, we will change VQA in the subtitle of Section 6.2 to document question answering.

---

### Meta-Review · Area_Chair_6GuR · 2023-09-14

**Recommendation:** 4

**Metareview:**

The paper introduces a foundational dataset that combines machine document comprehension with human eye movement data for visually-rich documents. Most reviewers have expressed considerable enthusiasm for this novel dataset, recognizing its significance in enhancing the comprehension of visually-rich documents, a notion substantiated by the performance improvements showcased in the paper's experiments.

However, the reviewers have raised pertinent issues that we encourage the authors to address:

1. The experimental procedure is notably time-intensive and demanding in terms of labor. It would be beneficial for the authors to explicitly acknowledge this as a limitation of their paper. Moreover, providing insights and recommendations for future research and researchers embarking on similar endeavors could be a valuable addition.

2. The issue of agreement among annotators warrants attention. While the paper elucidates the methodology employed to determine the final annotations, it is equally important to declare the level of agreement among the annotators. For instance, when instructing annotators to rate a sample on a scale of 1 to 5, it should be noted whether the average score of 3 results from all annotators assigning a score of 3 or if it arises from a split where half the annotators rate it as 5 and the other half as 1. Reporting such discrepancies in annotators' judgments is crucial.

3. To address the queries posed by the reviewers, it is advisable to consider incorporating additional experiments into the paper, as suggested by reviewer hWn3. Expanding the experimental scope could help provide comprehensive answers to these questions and further bolster the paper's contributions.

---

### Decision · Program_Chairs · 2023-10-07

**Decision:**

Accept-Findings

**Comment:**

The paper introduces a foundational dataset that combines machine document comprehension with human eye movement data for visually-rich documents. Most reviewers have expressed considerable enthusiasm for this novel dataset, recognizing its significance in enhancing the comprehension of visually-rich documents, a notion substantiated by the performance improvements showcased in the paper's experiments.

However, the reviewers have raised pertinent issues that we encourage the authors to address:

1. The experimental procedure is notably time-intensive and demanding in terms of labor. It would be beneficial for the authors to explicitly acknowledge this as a limitation of their paper. Moreover, providing insights and recommendations for future research and researchers embarking on similar endeavors could be a valuable addition.

2. The issue of agreement among annotators warrants attention. While the paper elucidates the methodology employed to determine the final annotations, it is equally important to declare the level of agreement among the annotators. For instance, when instructing annotators to rate a sample on a scale of 1 to 5, it should be noted whether the average score of 3 results from all annotators assigning a score of 3 or if it arises from a split where half the annotators rate it as 5 and the other half as 1. Reporting such discrepancies in annotators' judgments is crucial.

3. To address the queries posed by the reviewers, it is advisable to consider incorporating additional experiments into the paper, as suggested by reviewer hWn3. Expanding the experimental scope could help provide comprehensive answers to these questions and further bolster the paper's contributions.